# Nursing Intervention to Improve Positive Mental Health and Self-Care Skills in People with Chronic Physical Health Conditions

**DOI:** 10.3390/ijerph20010528

**Published:** 2022-12-28

**Authors:** Maria Aurelia Sánchez-Ortega, Maria Teresa Lluch-Canut, Juan Roldán-Merino, Zaida Agüera, Miguel Angel Hidalgo-Blanco, Antonio R. Moreno-Poyato, Jose Tinoco-Camarena, Carmen Moreno-Arroyo, Montserrat Puig-Llobet

**Affiliations:** 1Nursing and Occupational Therapy School (EUIT), Universitat Autònoma de Barcelona, 08221 Terrassa, Spain; 2Institut Català de la Salut (ICS), Generalitat de Catalunya, 08915 Barcelona, Spain; 3Departament d’Infermeria de Salut Pública, Salut Mental i Materno-Infantil, Escola d’Infermeria, Facultat de Medicina i Ciències de la Salut, Universitat de Barcelona (UB), 08907 L’Hospitalet de Llobregat, Spain; 4Campus Docent Sant Joan de Déu, Universitat de Barcelona, 08830 Sant Boi de Llobregat, Spain; 5Psychoneurobiology of Eating and Addictive Behaviors Group, Neurosciences Programme, Bellvitge Biomedical Research Institute (IDIBELL), 08908 L’Hospitalet de Llobregat, Spain; 6CIBER Fisiopatología de la Obesidad y Nutrición (CIBERObn), Instituto Salud Carlos III, 28015 Madrid, Spain; 7Departament d’Infermeria Fonamental i Médico-Quirúrgica, Escola d’Infermeria, Facultat de Medicina i Ciències de la Salut, Universitat de Barcelona (UB), 08907 L’Hospitalet de Llobregat, Spain; 8Center of Cornellà Specialists, Consorci Sanitari Integral, 08940 Cornellá de Llobregat, Spain

**Keywords:** intervention program, nursing intervention, self-care, self-care agency, positive mental health

## Abstract

The exponential increase in the number of people suffering chronic illness has become a problem for which healthcare services need a response. The inclusion of self-care and positive mental health as part of a strategy to promote health offers an opportunity for a reorganization oriented towards community spaces and group interventions. This study undertook the assessment of an intervention designed to optimize the agency of and capacity for self-care and positive mental health by utilizing activities drawn from the Nursing Intervention Classification (NIC), specifically from Field 3 (Behavioral), and organized as a program called PIPsE. A quasi-experimental design was prepared with an intervention group (n = 22) and a control group (n = 22), in a primary care center in the Barcelona metropolitan area. The instruments used were two ad hoc questionnaires to collect sociodemographic and satisfaction information and two scales: the Appraisal of Self-care Agency Scale (ASA) and the Positive Mental Health Questionnaire (PMHQ). The results obtained showed a significant increase in self-care capacity and both overall positive mental health and mental health by factors in the intervention group.

## 1. Introduction

The exponential increase in the number of people with chronic illnesses, directly related to increased life expectancy, has emerged as a problem of the first magnitude, producing a high demand for care which, far from appearing in an orderly manner, has presented itself as a great challenge for the coming years [1]. Many authors rightly place primary care as the motor for a healthcare model to cover this demand [2]. However, it is also evident from today’s news reporting that primary care is overwhelmed by the demands placed upon it, is lacking in resources, and is in real danger. Perhaps this is the moment to apply solutions that include self-care and positive mental health (PMH) as a strategy to promote self-responsibility in the population to become managers of their own personal and their family’s health [3,4], establishing an additional pillar in the system and thereby affording the chance to create clearer and more efficient procedural circuits in primary care.

The roots of the linking of the concepts of PMH with self-care capacity lie with the authors Orem and Vardiman, who identified 38 self-care behaviors susceptible to constituting positive mental health measures [5]. The idea is not a new one, but the step of putting it into practice in a systematic manner beyond simple advice and recommendations has not been taken before. We are speaking, then, of mobilizing resources from a perspective of psychosocial intervention in order to potentiate individual abilities that may be developed outside the setting of the primary care center, with the goal of reducing the overloading of the center. The proposal is one of interventions led by nurses and encompassing the community setting.

From the perspective of Orem, the self-care capacity takes form when individuals develop behavioral, cognitive, and emotional abilities to care for their own health [5,6,7]. All these abilities may be developed and yet remain inoperative—that is, for some reason the individual does not make use of them [8].

The construct of PMH is generating great interest internationally, as it contributes to the wellbeing of people [9,10,11,12,13,14,15,16,17]. The World Health Organization (WHO) defines it as “a state of mental wellbeing that enables people to cope with the stresses of life, realize their abilities, learn and work well, and contribute to their community.” The concept (and term) was first put forward by Marie Jahoda [18]; she argued that PMH was an individual and not a collective behavior, and that the social and cultural settings either facilitate or impede the achievement of health, and that this may vary over time. To make the construct operational, Lluch [19] structured it into six factors represented in thirty-nine items that make up the Multifactorial PMH model for assessment purposes [19].

Interventions have been proposed to promote PMH in nursing students [20,21] and informal caregivers working with people with chronic illnesses [22,23,24,25]. However, to our knowledge, no self-care and PMH based program conducted by nurses has been designed for addressing people with chronic problems using the Nursing Intervention Classification (NIC) taxonomy. Nursing interventions in this line may be useful to apply in primary care and could be adapted to others setting.

Considering the foregoing, we hypothesized that people with chronic diseases would benefit from a nursing intervention aimed at promoting PMH and self-care through psychosocial interventions based on the NIC. Following the intervention, we expected participants to have increased their levels of PMH and self-care agency. Therefore, the main aim was to assess the efficacy of a nursing psychosocial intervention program designed to increase the capacity for self-care and PMH. The specific goals of the study were to determine the level of satisfaction of the participants and to learn whether the proposed interventions were useful.

## 2. Materials and Methods

### 2.1. Design and Participants

A quasi-experimental design was established with an intervention group (IG; n = 22) and a control group (CG; n = 22). The sample was randomly selected from a prior participant list of those met the inclusion criteria: age 45 or above, chronic health problem(s), and acceptance of participation in the study. It was carried out in a publicly managed primary care center in the metropolitan area of Barcelona, Spain.

### 2.2. Assessment

Four instruments were used to collect data: (1) a sociodemographic data form with 8 variables (sex, age, marital status, number of children, education level, profession, work situation, and nationality); (2) the Appraisal of Self-care Agency Scale (ASA) Spanish validation [26], structured unidimensional into 24 items (16 positive and 8 negative) with questions 2, 6, 11, 13, 14, 15, 20, and 23 in reverse order to calculate the score, with four possible responses on a Likert-type scale with a cutoff of < 48 indicating low self-care (the lower the score, the less the capacity for self-care); (3) the PMHQ Positive Mental Health Questionnaire [19] based on the multifactorial model described above. The PMHQ has been validated in the Spanish population [27]. The PMHQ contains 39 items arranged unequally in reference to 6 factors—Factor 1: personal satisfaction (items 4, 6, 7, 12, 14, 31, 8, and 39); Factor 2: prosocial attitude (1, 3, 23, 25, and 37); Factor 3: self-control (2, 5, 21, 22, and 26); Factor 4: autonomy (10, 13, 19, 33, and 34); Factor 5: problem solving/self-maintenance (15, 16, 17, 27, 28, 29, 32, 35, and 36); and Factor 6: interpersonal relational skills (8, 9, 11, 18, 20, 24, and 30). Each item has four possible responses on a Likert-type scale. The questionnaire yields a global PMH score ranging from 39 to 156, with a higher score indicating greater positive mental health, as well as specific scores for each factor which vary according to the number of items per factor. The inverted negative items are 4, 23, 25, 37, 5, 21, 22, 26, 15, 16, 17, 27, 28, 29, 32, 35, 36, 11, 18, and 20 (ordered by factor); and 4) an ad hoc form made up of 10 questions to assess variables relating to the participants’ perception of the usefulness of the program, personal satisfaction with it, and overall assessment of the intervention. The instruments were administered at the outset and the conclusion of the program.

### 2.3. Procedure

In a first step, a list of patients who met the inclusion criteria and who were linked to the center was obtained. Recruitment was carried out with patients who consecutively attended this primary health care center. Patients were invited to participate on a voluntary basis. As the intervention groups had to be of 10–12 persons, control groups (randomized) were created with the same number of members. It was decided to carry out study with two intervention groups and two control groups of eleven subjects each.

Participants in the intervention group completed the sociodemographic and clinical form, the self-care agency scale, and the PMH questionnaire at baseline. Then, after participating in the intervention, they completed the self-care agency scale, the PMH questionnaire, and the satisfaction scale.

Participants in the control group completed the sociodemographic and clinical form, the self-care agency scale, and the PMH questionnaire at baseline. They did not participate in the intervention, but rather received standard follow-up. Afterwards, they completed the self-care agency scale and the PMH questionnaire.

### 2.4. Intervention

The Psychosocial Intervention Program for Nurses (PIPsE) is a working tool carried out by nurses to enhance patients’ own resources. The content of the program was based on the nursing intervention classification (NIC) from which five interventions were selected from Field 3—behavioral: maximizing socialization (NIC 5100), relaxation therapy (NIC 6040), simple directed imagination (NIC 6000), music therapy (NIC 4400), and education for health (NIC 5510). Each intervention was organized into activities designed to maximize some of the PMH factors as well as components of agency for self-care; taken together, they covered all the PMH factors and the components of self-care agency. The activities were organized along the following three lines: (1) activities aimed at managing communication skills, self-care, and self-management (with techniques such as breathing control, emotional regulation, relaxation, and music therapy); (2) activities of self-knowledge, autonomy, and prosocial attitude through personal reflection and sharing experiences with the group; and (3) activities to promote self-esteem using symbolic play, where the participation of each member was essential in order to achieve the objective.

The two IG (each n = 11) were conducted consecutively by two trained nurses, always the same in both groups. In total, it lasted four sessions of ninety minutes each on a weekly periodicity. Attendance reminders were made both in the previous session and telephonically two days before each session. Furthermore, an additional thirty minutes were set aside in the first and fourth sessions to fill out the questionnaires.

### 2.5. Statistical Analysis

The Fisher exact test was used for the comparison of categorical variables and the Mann–Whitney U test for the comparison of quantitative variables. To determine whether the intervention had produced a significant improvement in the intervention group, a multivariate analysis with covariance (MANOVA) was used to compare the pretest–posttest differences between the intervention and control groups.

Univariate analysis was also carried out with each pretest–posttest, using as the covariable the pretest score (ANCOVA). Finally, a variance analysis was made to determine whether the variation in the tests depended on age, marital status, level of education, work situation, number of children, use of analgesics, frequency of analgesic use, polymedication, or number of pathologies. Frequency and percentages were used to identify satisfaction and usefulness.

## 3. Results

The groups were homogeneous; their sociodemographic and clinical characteristics may be seen in Table 1. Statistically significant differences between the groups were only found for marital status (*p* = 0.015); the number of widowed people was greater in the intervention group.

At the outset of the study pretest, the mean PMH score for all of the participants was 116 (SD 13.7), and for self-care it was 63.7 (SD 6.0). Statistically significant differences were only found for PMH on F5 “Problem solving/self-maintenance” (*p* = 0.020); the patients from the intervention group had significantly higher scores than those from the control group (Table 2).

The results of the MANOVA with the pretest–posttest variables reveal statistically significant differences between the groups [F (7,36) = 28.51, *p* < 0.001].

The results of the ANOVA are presented in Table 3, where it will be observed that the intervention produced a significant increase in the PMH levels (both total score and for specific factors) and in the self-care agency. When we analyzed whether the variation was different for the different variables, for the MANOVA pretest–posttest this was not statistically significant; the intervention produced similar changes for the various age groups, the different marital status categories, the differing educational levels, work situations, number of children, use of analgesics, quantity of analgesics used, polymedication (or not), and number of pathologies. That is, the changes may be explained by the intervention program itself and not by other clinical or sociodemographic variables.

As may be seen in Table 4, the 22 participants in the intervention group answered that they would continue to do the activities that they had learned (100%), and 21 of them (95.5%) said they did so at home. All of the participants (100%) said that what they had learned in the intervention had been helpful in maintaining their health. In terms of whether there was improvement in self-perception related to the techniques learned, 86.4% answered in the affirmative, 9.1% said they needed more time, and 4.5% expressed doubt. As to the self-perception of the participants in assessing their capacity to carry out the exercises on their own, 86.4 % answered affirmatively while 13.6% responded in the negative. Regarding their satisfaction with the program, 100% of the participants said that they felt comfortable in their work group, would be inclined to repeat the experience, and would recommend it to others, and 90% gave the program a maximum score of 10 points, while the other 10% awarded it 9 points.

## 4. Discussion

The PIPsE intervention showed very satisfactory results, confirming our initial hypothesis. The results supported that those participants in the intervention group showed increased levels of PMH and self-care agency after the PIPsE program. Intriguingly, we also observed a decrease in these factors in the control group participants who did not participate in the intervention.

Regarding PMH, a significant increase was observed in all six factors that make up the construct, as well as in the global PMH score. These results are in line with previous studies that noted how specific interventions aimed at improving PMH can enhance specific factors such as “Personal satisfaction” and “Problem solving/self-care” in informal caregivers [22,23,24] or self-control in university nursing students [28].

The intervention group also obtained very satisfactory results in terms of self-care, with a statistically significant increase in self-care agency. At the same time, a decrease in self-care agency was observed in the control group. Comparing the results obtained for the two groups, nursing interventions designed to improve self-care agency in people with chronic health problems were effective, as in other studies [29,30]. Studies based on education have obtained positive results regarding improvement in patients’ participation in the process of their own self-care [31,32,33,34,35].

The intervention program was designed within a framework of psychosocial interventions [36]; of note there are other studies from the same milieu and with shared characteristics. Most of the programs we have found were designed to lower anxiety levels [37], and with good results. In this line, music therapy helps to boost mood and is commonly used in nursing intervention programs with good results [38]. Other studies aimed at the motivation and self-management of the chronic patient have confirmed that attention to and support of these patients are the keys to success [39,40].

Regarding the control group, it saw a reduction in the numbers for PMH and self-care agency. This is a reminder of the need to put forward nursing interventions for people with chronic health problems; intervention improves the capacity for self-care and PMH while the absence of intervention leads to a worsening of these. An important consideration in the type of intervention we are advocating here is that it may be easily and affordably implemented in any healthcare center.

### Limitations and Strengths

The present study has several limitations. First, it was carried out in a single center, so the sample may not be representative of the entire population of patients with chronic diseases. A second limitation of the study is the lack of long-term follow-up, which would have allowed us to learn whether the changes were maintained over time. The small sample size also limited the generalizability of the results. Further studies should address these limitations with larger samples and longer-term follow-up to provide greater robustness to the findings.

However, this study also presents the strength of applying a pioneering nursing intervention with multiple integrative activities based on the NIC. To our knowledge, no previous results have been described in the literature with PIPsE in the target population with chronic disease in the primary care setting.

## 5. Conclusions

The results suggest that the PIPsE program was effective in increasing self-care agency and improving PMH in the study population. These results highlight the need for intervention, as lack of intervention leads to a worsening in PMH and self-care agency in people with chronic physical health problems. In addition, participants in the intervention group showed a high level of satisfaction, both with the program in general and with the specific activities addressed in the sessions.

Due to the pressure on the healthcare system, especially in primary care, there is not enough time during healthcare visits to attend to the emotional needs of patients. In this vein, the group approach implies a decrease in consultation visits. In addition, an increase in PMH and self-care agency benefits people in their daily lives and allows them to face matters with greater internal resources, such as improved autonomy, personal satisfaction, social relationships, and quality of life, among others.

## Figures and Tables

**Table 1 ijerph-20-00528-t001:** Comparison of the sociodemographic and clinical characteristics of the sample.

	Total (n = 44)		Control Group (n = 22)		Intervention Group (n = 22)		
	Mean (SD)		Mean (SD)		Mean (SD)		*p*
Age	66.5 (9.7)		66.5 (9.7)		67.5 (9.8)		0.733
Number of pathologies	2.4 (1.6)		2.2 (1.6)		2.5 (1.5)		0.468
Number of children	2.2 (0.9)		2.2 (0.6)		2.2 (1.1)		0.902
	n	%	n	%	n	%	*p*
Age (intervals)							
45–55	7	15.9	5	22.7	2	9.1	0.385
56–65	10	22.7	4	18.2	6	27.3	
66–75	20	45.5	11	50.0	9	40.9	
>75	7	15.9	2	9.1	5	22.7	
Marital status							
Single	2	4.5	0	0.0	2	9.1	0.015
Married	34	77.3	21	95.5	13	59.1	
Widowed	7	15.9	1	4.5	6	27.3	
Divorced	1	2.3	0	0.0	1	4.5	
Education level							
None	21	47.7	12	54.5	9	40.9	0.547
Primary	22	50.0	10	45.5	12	54.5	
Vocational training	1	2.3	0	0.0	1	4.5	
Work situation							
Working	2	4.5	2	9.1	0	0.0	0.630
Unemployed	10	22.7	5	22.7	5	22.7	
Retired	32	72.7	15	68.2	17	77.3	
Analgesic use							
Yes	37	84.1	16	72.7	21	95.5	0.095
No	7	15.9	6	27.3	1	4.5	
Frequency of analgesic use							
Daily	17	45.9	7	43.9	10	47.6	0.999
Occasionally	27	54.1	9	56.3	11	52.4	
Polymedication							
Yes	15	34.1	7	31.8	8	36.4	0.999
No	29	65.9	15	68.2	14	63.6	

Mann–Whitney U test; Fisher exact test; *p*: significance level; SD: standard deviation.

**Table 2 ijerph-20-00528-t002:** Comparison of PMH and self-care at baseline.

	Total (n = 44)	Control Group (n = 22)	Intervention Group (n = 22)	*p*
	Mean (SD)	Mean (SD)	Mean (SD)	
Total Positive Mental Health	116.0 (3.7)	114.0 (13.2)	117.9 (14.2)	0.2851
F1: Personal satisfaction	23.0 4.5)	23.2 (4.9)	22.7 (4.1)	0.5091
F2: Prosocial attitude	17.0 (1.9)	17.1 (2.0)	17.0 (1.8)	0.7561
F3: Self-control	14.3 (2.7)	14.8 (2.5)	13.9 (2.9)	0.2881
F4: Autonomy	14.7 (1.9)	14.4 (1.4)	14.9 (2.4)	0.8481
F5: Problem solving/self-maintenance	26.3 (4.5)	24.7 (3.8)	28.0 (4.7)	0.0201
F6: Interpersonal relational skills	20.0 (3.3)	19.7 (2.7)	21.2 (3.7)	0.1821
Total self-care agency (ASA)	63.7 (6.0)	63.2 (5.2)	64.2 (6.9)	0.6131

Mann–Whitney U test; *p*: significance level; SD: standard deviation.

**Table 3 ijerph-20-00528-t003:** Pretest–posttest: differences in positive mental health and self-care agency.

	Intervention Group (n = 22)		Control Group (n = 22)		F Value (1.42)	*p*
	Mean	SD	Mean	SD		
Total Positive Mental Health	27.7	13.8	−6.5	7.2	105.3	<0.001
F1: Personal satisfaction	6.5	3.6	−2.1	2.2	89.6	<0.001
F2: Prosocial attitude	2.5	1.8	−0.6	0.8	57.1	<0.001
F3: Self-control	5.1	2.7	−1.1	1.4	92.2	<0.001
F4: Autonomy	2.5	2.2	−0.2	1.0	28.7	<0.001
F5: Problem solving/self-maintenance	6.1	4.6	−1.0	2.5	39.6	<0.001
F6: Interpersonal relational skills	4.7	2.9	−1.1	2.0	60.8	<0.001
Total self-care agency (ASA)	12.0	5.0	−2.2	4.4	99.9	<0.001

F value: pretest–posttest ANOVA. Mean, mean pre-test/posttest differences; SD, standard deviation.

**Table 4 ijerph-20-00528-t004:** Perceived satisfaction, usefulness, and learning vis-à-vis the PIPsE program.

	Intervention Group (n = 22)		Frequency
	YES	NO	% YES
Would carry out learned activities	22	0	100
Did the activities at home	21	1	95.5
Activities learned have been beneficial to health	22	0	100
Self-perception of learning activities	19	3	86.4
Need more time for learning	2	20	9.1
Doubts about whether more time is needed for learning	1	21	4.5
Ability to carry out exercises on their own	19	3	86.4
Feel comfortable in the working group	22	0	100
Would repeat and recommend the experience	22	0	100

## Data Availability

Data available on request from the authors.

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
