# Peer review of "Nursing Intervention to Improve Positive Mental Health and Self-Care Skills in People with Chronic Physical Health Conditions"

_ijerph, 2022, doi:10.3390/ijerph20010528_

Round 1
Reviewer 1 Report
1- The text is written in a very complicated manner. The sentences are very long; and a full English proof reading is required.
2- How do you justify such low number of participants in a quantitative method? What was the sampling process?
(Sample size is a notable limitation that significantly affect generalization of the findings.)
3- I see the same method, same questions, and same results comparing to the study that the same authors did in 2012 in the same center and published (link can be found in the below line). How do you explain the novelty of the present research?
(https://bmcpublichealth.biomedcentral.com/articles/10.1186/1471-2458-13-928)
Author Response
#Reviewer 1
We deeply appreciate the reviewer's comments and believe that they substantially improve the manuscript. The current version of the manuscript includes modifications in response to the concerns raised by the reviewer and some additional changes, which we think could improve the article significantly. Our responses to the reviewer can be seen below:
1- The text is written in a very complicated manner. The sentences are very long; and a full English proof reading is required.
Reply: As suggested by the reviewer, the entire manuscript has been revised by a native English speaker.
2- How do you justify such low number of participants in a quantitative method? What was the sampling process? (Sample size is a notable limitation that significantly affect generalization of the findings.)
Reply: We understand the reviewer's concern. Therefore, the following explanation has been included in the Method section: In a first step, a list of patients who met the inclusion criteria and who were linked to the center was obtained. Recruitment was carried out with patients who consecutively attended this primary health care center. Patients were invited to participate on a voluntary basis. As the intervention groups had to be of 10-12 persons, control groups (randomized) were created with the same number of members. It was decided to carry out a study with two intervention groups and two control groups of eleven subjects each.
In addition, the reviewer is right in pointing out that the small sample size may limit the generalizability of the results. Because this point has been noted in the limitations section. However, we decided to present the findings of this study because we consider the results to be sufficiently novel and convincing. The study has the strength of applying a pioneering intervention, with activities taken from the NIC (Nursing Interventions Classification) that, to the best of our knowledge, there is no record of previous results in the literature. All this has now been noted in the limitations and strengths section.
3- I see the same method, same questions, and same results comparing to the study that the same authors did in 2012 in the same center and published (link can be found in the below line). How do you explain the novelty of the present research?
(https://bmcpublichealth.biomedcentral.com/articles/10.1186/1471-2458-13-928)
Reply: the reviewer is correct in stating that there might be some similarities with the previous article referred to (Lluch-Canut et al.; 2013) (e.g., assessment, recruitment). However, the present study is the second phase of the previous one and, thus, the design, objectives and results of this study are completely different. More specifically, the previous study was conducted with a larger sample (259 adults with chronic diseases), but we used a cross-sectional design, not pre-post as the present one. In the previous study we wanted to provide a description of PMH levels in this population and their correlations with clinical and sociodemographic factors. However, in the present study the purpose is to expand knowledge by analyzing the impact of the PIPsE intervention and the changes that this completely novel specific intervention produces in PMH and self-care agency in this target population.
Reviewer 2 Report
Dear authors, "This study aimed to assess the effect of a positive mental health nursing intervention in people with chronic physical health conditions". It is an honor for me to review this article.
1. The introduction can be improved with more relevant and recent references. In the last 5 years, many scientific and relevant studies have been published on self-care abilities and positive mental health in people with chronic diseases. Please, I invite you to use Pubmed to find out about recently published studies on the subject.
It is convenient to include the study hypothesis and link it to the results.
2. In the method section it is recommended to include an explanation of the procedure for calculating and selecting the sample. As well as the integration process of the study groups.
3. Expand the description of the intervention. Who made the delivery?
How was its fidelity maintained?
4. Refine the conclusion with more suggestions about the future implications of the study and the theoretical implications.
Author Response
#Reviewer 2
We deeply appreciate the reviewer's comments and believe that they substantially improve the manuscript. The current version of the manuscript includes modifications in response to the concerns raised by the reviewer and some additional changes, which we think could improve the article significantly. Our responses to the reviewer can be seen below:
1- The introduction can be improved with more relevant and recent references. In the last 5 years, many scientific and relevant studies have been published on self-care abilities and positive mental health in people with chronic diseases. Please, I invite you to use Pubmed to find out about recently published studies on the subject.
Reply: As suggested by the reviewer, we have added new and more updated references on the topic of the study.
1b- It is convenient to include the study hypothesis and link it to the results.
Reply: In agreement with this suggestion, the hypothesis of the study has been included and discussed according to the results.
- In the method section it is recommended to include an explanation of the procedure for calculating and selecting the sample. As well as the integration process of the study groups.
Reply: We fully understand the reviewer’s concern. Therefore, the following explanation has been included in the Method section: In a first step, a list of patients who met the inclusion criteria and who were linked to the center was obtained. Recruitment was carried out with patients who consecutively attended this primary health care center. Patients were invited to participate on a voluntary basis. As the intervention groups had to be of 10-12 persons, control groups (randomized) were created with the same number of members. It was decided to carry out a study with two intervention groups and two control groups of eleven subjects each.
We are aware that the small sample size may limit the generalizability of the results, and this has been noted in limitations section. However, we decided to present the findings of this study with two intervention groups and two control groups because we consider the results to be sufficiently novel and convincing. The study has the strength of applying a pioneering intervention, with activities taken from the NIC (Nursing Interventions Classification) that, to the best of our knowledge, there is no record of previous results in the literature. All this has also been noted in the limitations and strengths section in the Discussion.
- Expand the description of the intervention. Who made the delivery? How was its fidelity maintained?
Reply: We fully agree with this suggestion. Therefore, we have included the following description of the PIPsE intervention:
“The activities were organized along the following three lines: 1) activities aimed at managing communication skills, self-care, and self-management (with techniques such as breathing control, emotional regulation, relaxation, and music therapy); 2) activities of self-knowledge, autonomy and prosocial attitude through personal reflection and sharing experiences with the group; and 3) activities to promote self-esteem using symbolic play, where the participation of each member was essential to achieve the objective.
The two IG (each n=11) were conducted consecutively by two trained nurses, al-ways the same in both groups. In total, it lasted four sessions of ninety minutes each, on a weekly periodicity. Attendance reminders were made both in the previous session and telephonically two days before each session”
- Refine the conclusion with more suggestions about the future implications of the study and the theoretical implications.
Reply: Thank you for pointing this out. As suggested by the reviewer, the Conclusions section has been to be rewritten as follows:
“The results suggest that the PIPsE program was effective in increasing self-care agency and improving PMH in the study population. These results highlight the need for intervention, as lack of intervention leads to a worsening in PMH and self-care agency in people with chronic physical health problems. In addition, participants in the intervention group showed a high level of satisfaction, both with the program in general and with the specific activities addressed in the sessions.
Because of the pressure of the healthcare system, especially in primary care, there is not enough time during healthcare visits to attend to the emotional needs of patients. In this vein, the group approach implies a decrease in consultation visits. In addition, an increase in PMH and self-care agency benefits people in their daily lives and allows them to face matters with greater internal resources, such as improved autonomy, personal satisfaction, social relationships, and quality of life, among others.”
Round 2
Reviewer 1 Report
1- Thanks for the explanations and corrections.
2- Please add the "data collection date (year)" in the Method section.